# Analysis and Risk Assessment of Pesticide Residues in Strawberry Using the PRIMo Model: Detection, Public Health and Safety Implications

**DOI:** 10.3390/foods14203470

**Published:** 2025-10-11

**Authors:** Elvira De Rosa, Maddalena Di Lillo, Maria Triassi, Fabiana Di Duca, Immacolata Russo, Vito Graziano, Giovanni Mazzei, Immanuela Gentile, Seyedeh Zahra Shojaeian, Paolo Montuori

**Affiliations:** 1Department of Public Health, “Federico II” University, Via Sergio Pansini nº 5, 80131 Naples, Italy; derosaelvira92@gmail.com (E.D.R.);; 2Department of Human Sciences and Quality of Life Promotion, San Raffaele University, 00166 Rome, Italy; 3Department of Public Health, “Federico II” University Hospital, Via Sergio Pansini nº 5, 80131 Naples, Italy; imrusso@unina.it

**Keywords:** strawberries analysis, multiresidues, QuEChERS method, EFSA PRIMo model, cumulative dietary risk

## Abstract

Strawberries are among the most consumed fruits in Europe, but intensive cultivation requires frequent pesticide use, raising food safety concerns. This study evaluated pesticide residues and dietary risk in strawberries from the Agro Aversano area (Southern Italy). A total of 83 samples collected in 2023–2024 were analyzed using a validated QuEChERS-LC–MS/MS method targeting 850 active substances. Thirty-one pesticides were detected, predominantly fungicides, followed by acaricides and insecticides. Cyflumetofen and pyrimethanil were the most frequent residues, while compounds with low toxicological thresholds, including emamectin benzoate, lambda-cyhalothrin, acetamiprid, and tetraconazole, were also identified. Dietary risk assessment was conducted with the EFSA PRIMo model (v.3.1), focusing on the NL toddler subgroup. Despite occasional exceedances of European Maximum Residue Limits (MRLs), both acute and chronic exposures remained well below toxicological reference values (ADI and ARfD). Overall, the results indicate negligible health risks but highlight the relevance of cumulative exposure, underscoring the need for continuous monitoring and sustainable pest management practices to ensure strawberry safety.

## 1. Introduction

Strawberry production has seen significant global growth, surpassing 9 million tonnes [1]. Italy ranks 14th worldwide, with over 119,000 tonnes produced annually, primarily in southern regions, which account for more than 60% of national output [2,3,4]. The Mediterranean basin, including Spain, Italy, Greece, Morocco, Egypt, and Turkey are the main supplier for the European market from winter to late spring [2]. Despite rising prices, strawberries remain one of the most consumed seasonal fruits in Italy, purchased by 74% of families and representing 6.7% of their fruit expenditure [5,6,7].

However, climate change is increasingly impacting strawberry cultivation. A U.S. study showed that a 3 °F rise can reduce yields by 40% [8]. Recent atypical weather conditions in the Mediterranean low temperatures, heavy rainfall, and temperature fluctuations have slowed production and promoted the spread of pathogens. High humidity favors *Botrytis cinerea*, while water stress supports *Sphaerotheca macularis* development. Warmer temperatures also accelerate the spread of *Tetranychus urticae*, *Frankliniella occidentalis*, and aphids, which cause direct damage and transmit viruses. Milder winters have enabled earlier infestations by *Drosophila suzukii* and increased survival of *Spodoptera littoralis* [9].

To manage these threats, various pesticides are applied, including insecticides, fungicides, and acaricides. Fungicides such as SDHIs (Succinate Dehydrogenase Inhibitors) are linked to neurological and cancer-related effects [10], QoIs (Quinone outside Inhibitors) may induce neurotoxicity [11], and triazoles though individually less harmful can cause endocrine disruption when combined [12,13]. Insecticides include acetylcholinesterase inhibitors [14,15], pyrethroids [14,16,17], and newer compounds like neonicotinoids, spinosyns, diamides, and spirotetramat, which target specific neural or metabolic pathways [18]. Acaricides commonly inhibit mitochondrial complexes, e.g., fenpyroximate, tebufenpyrad, cyflumetofen, and bifenazate.

Strawberries may thus carry multiple pesticide residues. The “multi-residue” phenomenon simultaneous presence of several active substances raises concern due to additive, synergistic, or antagonistic effects that may enhance toxicity. Chronic exposure to these substances has been associated with gut microbiota alterations, liver damage, cognitive decline, and metabolic disorders [12,13], as well as increased risks of breast [19] and colon cancer [20]. Washing and removing strawberry caps can help reduce residue levels [21,22].

Despite these concerns, strawberries are widely appreciated not only for their taste but also for their health benefits [7,23]. They are rich in vitamin C, folates, flavonoids, anthocyanins, and phenolic acids, which exert antioxidant and anti-inflammatory effects. These compounds support DNA repair, regulate gene expression, and may protect against chronic diseases like cardiovascular conditions, cancer, type II diabetes, and neurodegenerative disorders [24,25]. Clinical studies have also shown positive effects in reducing dyslipidemia, oxidative stress, and inflammatory cytokines in metabolic syndrome [24,26,27]. In light of the potential duality between health benefits and chemical risks, this study aims to detect pesticide residues in strawberries at harvest and evaluate the associated health risk. The risk assessment was performed using PRIMo (Pesticide Residue Intake Model), the dietary exposure tool developed by EFSA (European Food Safety Authority) for estimating chronic and acute intake of pesticide residues based on real consumption data from European populations. This model allows for a reliable and standardized estimation of potential consumer exposure, ensuring that the results have strong implications for public health and food safety policies.

The novelty of this study lies in its focus on strawberries produced in the Agro Aversano area, one of the most intensive agricultural districts in Southern Italy, where high anthropogenic pressure and intensive farming practices may increase the risk of pesticide contamination. By integrating residue analysis with toxicological reference values and local consumption data, this research provides new insights into cumulative dietary exposure and its relevance for both consumer protection and the promotion of sustainable agricultural practices.

## 2. Materials and Methods

### 2.1. Study Area

The study was conducted in the Campanian Plain (Figure 1), a vast and fertile agricultural Region in southern Italy, renowned for its intensive farming activities. This area benefits from a Mediterranean climate, characterized by mild winters and hot, dry summers, providing optimal conditions for the cultivation of various crops, including strawberries [28]. Particular focus was given to the Agro Aversano; a sub-region of the Campanian Plain located in the province of Caserta. This area has historically been recognized for its high agricultural productivity, especially in fruit and vegetable cultivation. However, intensive farming practices often involve the widespread use of pesticides to maximize yields and protect crops from pests and diseases.

In this context, strawberry samples were collected from 21 farms distributed across the Agro Aversano, ensuring adequate representativeness of the main production sites within the area. The sampling was performed over the course of the strawberry growing season (from spring to early summer), thereby accounting for the temporal variability in pesticide applications typically associated with different phenological stages of the crop. This approach was designed to provide a comprehensive overview of residue levels and potential dietary exposure risks. The Agro Aversano was selected not only due to its significant role in strawberry production but also for its relevance to both the local and national agricultural economy.

### 2.2. Sample Preparation

From November to June 2023 and 2024, 83 strawberry samples were collected from multiple strawberry growers in Agro-Aversano area. As defined in Commission Directive 2002/63 EC [29], a minimum concentration of each sample was 1 kg.

The strawberry samples collected were fresh and not washed prior to the analytical phase to avoid compromising the results, as washing could have reduced the pesticide concentration in the samples. To ensure the representativeness of the results, all samples were homogenized using a homogenizer. The samples were then stored in a refrigerator at temperatures between 4 and 8 °C until they were subjected to analytical determination using the QuEChERS method.

The sample preparation and analysis were conducted using the UNI EN 15662 Modular QuEChERS method; as illustrated in Appendix A, the ionic peaks of the pesticides that were analyzed are reported (Appendix A) [30].

For the initial extraction step, 10.0 g of homogenized strawberry sample were accurately weighed into a 50 mL polypropylene conical centrifuge tube using an analytical balance. To each tube, 10 mL of acetonitrile (ACN, HPLC-grade) were added. The mixture was vortexed (Vortex GENIE 2, Scientific Industries, New York, NY, USA) for exactly 1 min at maximum speed to ensure thorough extraction of analytes into the organic phase.

In the second phase, a QuEChERS salt mixture consisting of 4.0 g anhydrous magnesium sulfate (MgSO_4_), 1.0 g sodium chloride (NaCl), 1.0 g trisodium citrate dihydrate, and 0.5 g disodium hydrogen citrate sesquihydrate was added to each tube. Tubes were immediately vortexed for 2 min to promote efficient partitioning of pesticides into the ACN phase and then centrifuged at 6000 rpm for 5 min at 4 °C to facilitate phase separation.

For the dispersive solid-phase extraction (d-SPE) cleanup, 4.0 mL of the upper ACN layer (supernatant) were transferred into a 15 mL polypropylene tube prefilled with 600 mg anhydrous MgSO_4_ and 200 mg primary secondary amine (PSA). MgSO_4_ was used to remove residual water, while PSA eliminated polar interferences such as sugars, organic acids, and fatty compounds. Tubes were vortexed for 1 min and centrifuged again at 6000 rpm for 5 min.

The resulting supernatant (approximately 3.5–3.8 mL) was carefully transferred into amber glass vials to minimize photodegradation and acidified by adding 20 µL of formic acid (≥98%, LC-MS grade) per mL of extract, improving the stability of base-sensitive pesticides during storage. The final extracts were stored at −20 °C until instrumental analysis.

### 2.3. Instrumental Analysis

Instrumental analysis was performed on an Agilent 6495C, (Agilent Technologies, Santa Clara, CA, USA) triple quadrupole LC/MS. Chromatographic separation was performed on an Acclaim C18 column (4 μm particle size, 250 × 4.6 mm). The mobile phase consisted of water/methanol (95:5) (LC-MS grade) for eluent A, while eluent B contained methanol/water (95:5) (LC-MS grade); each of them contains 0.1% formic acid/5 mM ammonium formate. By changing the ratio of solvent A to solvent B, a gradient elution was executed: 5% at 0 min; 95% at 10 min; 95% at 13 min; and 5% at 13.01 min. Separation was achieved at 25 °C using a flow rate of 0.3 mL min^−1^ and the injection volume of 2 μL. Ionization and fragmentation settings were optimized by direct injection of pesticides standard solutions. MS/MS was performed in the selected reaction monitoring mode (SRM) with electrospray ionization (ESI) in positive mode. Mass Hunter software (Applied Biosystems, Waltham, MA, USA) was used for data acquisition, processing for analyte confirmation, and quantitative analysis. The flow rate parameters applied in this study included a capillary voltage of 4000 V, a fragmentor voltage of 190 V, a drying gas flow rate of 9 L/min, and a drying gas temperature of 325 °C. Additionally, the collision gas was set to medium, while the nebulizer gas and auxiliary gas pressures were maintained at 45 psi (Appendix A).

### 2.4. Quality Assurance and Control (QA/QC)

The analytical method was validated following the criteria set by SANTE/11312/2021 [31]. Following the guidelines mentioned above, the method validation focused on evaluating performance parameters such as the working range with linearity, detection thresholds (LODs), quantification thresholds (LOQs), precision, and reproducibility. The linearity criterion was considered fulfilled when the correlation coefficient (R^2^) was ≥0.990. For recovery assessments, both precision and reproducibility were regarded as acceptable if they fell within the 80–120% range (Appendix A). Linearity was assessed by spiking a blank matrix with target analytes at six concentrations (0.005–0.25 mg kg^−1^) and constructing matrix-matched calibration curves. Accuracy was evaluated in recovery experiments using three replicates at two fortification levels (0.01 and 0.05 mg kg^−1^). The limit of detection (LOD) was determined as the lowest analyte concentration yielding a signal-to-noise ratio of 3:1. The LOQs (0.010 mg kg^−1^) were determined as ten times the standard deviation of replicate analyses of fortified blank samples. Precision was expressed as relative standard deviations (%RSDs) obtained from five replicates on the same day (repeatability) and across three consecutive days (reproducibility).

### 2.5. Dietary Exposure Risk Assessment Using the PRIMo Model Version 3.1

For the assessment of dietary exposure to pesticides, the EFSA PRIMo version 3.1, was used. This tool was developed by EFSA to estimate pesticide residue intake in the European population, considering both chronic (long-term) and acute (short-term) exposure scenarios. The PRIMo model supports risk assessments in a transparent manner, reflecting the risk assessment approach currently agreed upon at European Union (EU) level [29]. It enables standardized risk assessments in line with EU regulatory requirements. Version 3.1 has been updated with new food consumption data to improve the accuracy of dietary exposure estimates for pesticide residues. Risk assessment considers two exposure timeframes:-Acute exposure, estimated through the IESTI (International Estimated Short-Term Intake) and compared to the ARfD (Acute Reference Dose), defined as the amount of a substance that can be ingested over a short period (e.g., one meal or one day) without appreciable health risk.-Chronic exposure, estimated through the NEDI (National Estimated Daily Intake) and compared to the ADI (Acceptable Daily Intake), which represents the amount of a chemical substance that can be ingested daily over a lifetime without appreciable health risk.

The model integrates the following elements: Age and country specific food consumption data (e.g., “NL toddler” for young children in the Netherlands), residue levels in food commodities, including the Supervised Trials Median Residue (STMR) for chronic risk assessment and the Highest Residue (HR) for acute risk assessment, toxicological reference values, such as the ADI and the ARfD.

This allows for comparison of estimated exposure levels against established safety thresholds, identifying potential exceedances or safety margins, with particular attention to vulnerable groups, such as children. If exposure levels are below the toxicological reference values, it can be concluded that consumer exposure does not pose an unacceptable health risk.

PRIMo uses food consumption data collected through standardized national dietary surveys (EU Menu), stratified by age, sex, and country. In this study, the NL Toddler group was selected, representing the 8–20 months age class, considered among the most sensitive subpopulations in terms of exposure per kg body weight, with a reference body weight of 10.20 kg [32,33].

#### 2.5.1. Acute Exposure Assessment Based on IESTI Estimation

As part of the PRIMo (version 3.1), the IESTI was used to assess acute dietary exposure to pesticides, in accordance with FAO/WHO guidelines [34] and following the calculation methodology adopted by the European Food Safety Authority [29].

The IESTI represents the estimated amount of a pesticide residue that can be ingested in a single meal or within a short period (typically one day), based on the acute consumption of a specific food commodity. It is used to evaluate potential acute health risks from the intake of pesticide-contaminated food by comparing the estimated exposure to the ARfD, a toxicological threshold derived from acute or subacute toxicity studies.

The ARfD is expressed in mg kg^−1^ body weight and is established using the NOAEL (No Observed Adverse Effect Level) or LOAEL (Lowest Observed Adverse Effect Level) with the application of uncertainty factors similar to those used for the ADI. An ARfD is not defined for all active substances; it is only established when sufficient scientific evidence exists regarding the acute toxicity of a single dose. The IESTI is calculated using one of three standard equations, depending on the unit weight of the food commodity and the nature of the produce [35]:Case 1: unit weight <25 g (e.g., strawberries, cherries),Case 2a: unit weight >25 g but smaller than the portion consumed (e.g., mandarins, peaches),Case 2b: unit weight >25 g and larger than the portion consumed (e.g., melons, cabbages).

Calculation parameters are based on consumption and residue data provided by various EU Member States. In the present study, since the matrix under investigation—strawberries—has a unit weight below 25 g, Case 1 was applied. The corresponding IESTI formula is:IESTI Case 1: LP × HR × PF × CF/BW(1)
where

LP = Large Portion, the high-end daily consumption (kg) for the target population subgroup;HR = Highest Residue level detected in the commodity (mg kg^−1^);PF = Processing factor, accounting for changes in residue levels due to food preparation or peeling;CF = Conversion factor (applied if relevant, otherwise =1);BW = Body weight (kg) of the target population subgroup.

Once the IESTI is calculated, it is compared with the ARfD to determine the level of risk, expressed as a percentage:%ArfD = IESTI × 100/ARfD(2)

An acute dietary risk is considered acceptable when %ARfD < 100%, and unacceptable when %ARfD > 100% [36]. In accordance with standard methodology, the Highest Residue (HR) value detected in the analytical samples was used to simulate a worst-case acute exposure scenario.

#### 2.5.2. Chronic Exposure Assessment Based on NEDI Estimation

The PRIMo estimates chronic dietary exposure to pesticides based on the NEDI, in accordance with FAO/WHO guidelines and using the calculation approach adopted by the EFSA [34]. The NEDI represents the long-term daily intake of a pesticide residue by the population through the habitual consumption of contaminated foods. It is used to evaluate the chronic health risk by comparing the estimated exposure to the ADI [37].

The ADI is a toxicological reference value that indicates the amount of a substance present in food or drinking water that can be ingested daily over a lifetime without appreciable health risk. It is derived from toxicity studies in laboratory animals (typically rats, mice, or dogs) and is based on the NOAEL, which is the highest dose at which no adverse effects are observed. To ensure a high margin of safety, a standard uncertainty factor of 100 is applied, accounting for both interspecies (animal to human) and intraspecies (human variability) differences.

The ADI is expressed in mg kg^−1^ body weight per day and is established for all active substances, including those without acute toxicity, to account for potential cumulative or long-term effects relevant to food safety. Chronic dietary risk was assessed using the following equation:NEDI = APR × *average daily consumption*/1000(3)
where

-APR (Average Pesticide Residue) = mean concentration of pesticide residues (mg kg^−1^) detected in analyzed samples;-Average Daily Consumption = average intake of the food commodity (g/kg bw/day); for the NL toddler population group (8–20 months), this value was 0.344 g/kg bw/day;-1000 = conversion factor to harmonize units (from grams to kilograms).

The NEDI was then compared to the ADI to determine the level of chronic risk, expressed as a percentage:%ADI = NEDI × 100/ADI(4)

Chronic exposure was considered acceptable when %ADI < 100% and unacceptable when %ADI ≥ 100%. Unlike the standard EFSA PRIMo approach, which typically relies on STMR values from controlled field trials, this study used average residue levels (APR) directly calculated from analytical data obtained from strawberry samples. This methodological choice aimed to more accurately reflect realistic consumer exposure within the specific geographical context of the study. Although not harmonized at the regulatory level, this approach enables a more localized and representative assessment of chronic dietary risk associated with the consumption of contaminated food commodities [38].

## 3. Results

### 3.1. Results of Multiresidue Pesticide Analysis in Strawberries

In the present study, a total of 83 strawberry samples were analyzed, all collected directly from the field within the Agro Aversano study area. Each strawberry producer, and consequently each field where sampling was carried out, was assigned an identification code consisting of a letter of the alphabet and a progressive number (from A1 to Z2). This procedure ensured the confidentiality of the producers. A multiresidue analytical approach was employed, covering 850 active substances belonging to the main classes of both authorized and unauthorized pesticides under European regulations. The results revealed the presence of 31 different active substances detected in at least one sample. The data, shown in Appendix A, illustrate the frequency with which each pesticide was identified in the analyzed samples. Cyflumetofen was by far the most frequently detected compound (25 samples), nearly twice as often as tetraconazole, the second most frequently found (13 samples), followed by bupirimate (12 samples). As shown in Table 1, fungicides emerged as the most represented category, both in terms of the number of active substances detected and their overall concentrations. Although detected in only 6 samples, pyrimethanil exhibited the highest concentrations, with a total concentration of 3.806 mg kg^−1^, a mean concentration of 0.634 mg kg^−1^, and a maximum concentration of 2.000 mg kg^−1^, observed in sample I5. This was followed by the acaricide cyflumetofen, found in 25 samples, which had a total concentration of 2.803 mg kg^−1^ but a low mean concentration of 0.112 mg kg^−1^; the fungicide fluxapyroxad, present in 8 samples, had a total concentration of 2.138 mg kg^−1^ and a mean of 0.267 mg kg^−1^; and the fungicide bupirimate had a maximum concentration of 1.508 mg kg^−1^ and a mean of 0.126 mg kg^−1^. All other pesticides showed total concentrations below 1 mg kg^−1^, despite the detection of some substances such as tetraconazole and fluopyram in 13 and 12 samples, respectively. The lowest detected concentration of 0.1 mg kg^−1^ was observed for several substances, including emamectin, trifloxystrobin, bifenazate, epirimol, difenoconazole, hexythiazox, and chlorantraniliprole. The latter, an insecticide, also exhibited the lowest mean concentration of 0.011 mg kg^−1^.

Figure 2 illustrates the percentage distribution of pesticide concentrations detected in strawberry samples from the Agro Aversano area, broken down by pesticide category and by individual active substances, respectively. Fungicides clearly predominated, accounting for 66% of the total pesticide concentration. Among these, pyrimethanil (21.60%) and fluxapyroxad (12.14%) were the most represented, suggesting their significant use in strawberry cultivation. The acaricide category, though comprising only five active substances, represented the second largest share (22%), with cyflumetofen alone contributing 15.91% to this category. Insecticides accounted for 12% of the total concentration, showing lower overall levels compared to the other two main categories, despite being more numerous and including spinosad, which was detected in 9 samples but at very low concentrations (total concentration of 0.273 mg kg^−1^ and mean of 0.030 mg kg^−1^). Among insecticides, flupyradifurone was the most relevant (4.43%). Other active substances such as bupirimate (8.56%), fluopyram (4.09%), fenhexamid (2.82%), difenoconazole (3.27%), cyprodinil (1.48%), and boscalid (1.68%) contributed smaller percentages to the total, reflecting a diversified use of chemical compounds for strawberry protection.

Figure 3 presents the distribution of samples according to the number of pesticide residues detected. This analysis is essential to assess the extent of the multiresidue issue, namely the simultaneous presence of multiple chemical substances in a single sample. Twenty-eight percent of the samples contained a single active substance, most commonly cyflumetofen (7 samples) and bifenazate (5 samples); 17% of the samples were residue-free, while the remaining 55% contained between 2 and 6 residues (as in sample I4), and up to 7 residues, detected in sample D2.

The multiresidue analysis conducted on a wide range of strawberry samples collected in the Agro Aversano area between January 2023 and June 2024 provided significant results from food safety, agronomic, and environmental perspectives. As noted previously, numerous phytopathological issues can compromise crop productivity, which explains the presence of various active substances identified through the multiresidue screening, aimed at combating different pathogens [39]. The results revealed a significant presence of fungicides and acaricides in strawberries from the Agro Aversano area, reflecting the specific phytosanitary needs of this crop. The rising temperatures in recent years have contributed to increased fungal development in strawberry cultivation, which is known to be highly susceptible to several fungal diseases, particularly powdery mildew (*Sphaerotheca macularis*) and gray mold (*Botrytis cinerea*), both of which can severely affect yield and fruit quality. Consequently, fungicides represented the most frequently detected pesticide category (66%). Among the most frequently used active substances were tetraconazole (IBE), bupirimate, and fluopyram in terms of detection frequency, and pyrimethanil and fluxapyroxad in terms of concentration. Other detected fungicides included penconazole (IBE), fenhexamid, boscalid (SDHI) often applied in combination with pyraclostrobin (QOI), which degrades so rapidly that it did not appear in the analysis. The absence or low detection of certain substances can be attributed to a combination of factors, such as the selectivity of phytosanitary treatments, the physicochemical properties of the molecules (persistence, solubility, pre-harvest intervals), and local microclimatic conditions that may favor the natural degradation of residues. Residues of cyflufenamid, fludioxonil, penthiopyrad, cyprodinil, captan, and azoxystrobin (QOI) were occasionally observed. In some cases, the presence of two fungicides was the result of mixed formulations, such as trifloxystrobin (QOI), which was consistently found in combination with fluopyram (SDHI) in the nine relevant samples, or difenoconazole (IBE), which co-occurred with fluxapyroxad (SDHI) in the five samples where it was detected. This highlights the issue of multiresidue presence, found in more than half of the analyzed samples (55%). It is important to note that even when individual residues are below regulatory limits, cumulative exposure may still pose a risk, particularly for vulnerable populations (e.g., children, pregnant women), due to potential synergistic or additive effects, many of which remain insufficiently studied. In sample E5, the co-occurrence of tetraconazole and fenhexamid, both triazole fungicides, raises concerns regarding cumulative toxicological effects, as their combined exposure may interfere with xenobiotic metabolism and hormone synthesis, thereby increasing potential health risks.

Special attention should be given to cyflumetofen, an acaricide belonging to the methylphenylacetate class, used to control phytophagous mites such as the two-spotted spider mite (*Tetranychus urticae*), whose proliferation is favored by changing climatic conditions. The multiresidue analysis revealed both a high frequency of detection (25 samples) and a substantial total concentration. Its effectiveness is likely due to its recent introduction and the limited resistance developed by target pests. However, the observed variability in concentration may suggest non-compliant or excessive use of the substance, raising concerns about proper pre-harvest interval management and the efficacy of monitoring systems. Other acaricides such as hexythiazox, tebufenpyrad, and bifenazate were sporadically detected at negligible concentrations, while fenpyroximate was found only in sample M3.

Although insecticides were detected less frequently, their consistent presence indicates the need to manage phytophagous insects that affect strawberry crops. Among these are aphid species (*Macrosiphum euphorbiae*, *Sitobion fragariae*, *Aphis gossypii*, *Chaetosiphon fragaefolii*), which are targeted with flupyradifurone having the highest concentration among insecticides (0.780 mg kg^−1^) as well as pirimicarb, acetamiprid, and lambda-cyhalothrin, the latter found only in sample A7. Spinosad, detected in nine samples, is widely used against both noctuid moths (*Spodoptera littoralis*) and thrips (*Frankliniella occidentalis*). Other substances used against noctuids include emamectin benzoate and chlorantraniliprole, both currently being phased out, while spinetoram is used for thrip control. For the management of the spotted wing drosophila (*Drosophila suzukii*), pheromone traps are preferred over insecticides, especially toward the end of the strawberry season, to avoid increased residue levels.

As shown with fungicides, the data reveal that a significant portion of samples contain multiple residues (55%); in some cases, up to seven different active substances were detected in a single sample. Although these residues fall within legal limits, this multiresidue phenomenon raises important toxicological considerations. Current regulations establish limits for each individual active substance, but the scientific community and regulatory bodies are increasingly focusing on the potential combined or synergistic effects of simultaneous exposure to multiple residues, particularly for vulnerable consumers such as children and the elderly. Appendix A showed a representative chromatogram of the pesticides analyzed in strawberry samples and the mass spectrum of Acetamiprid was also shown.

### 3.2. Health Risk Assessment Results Related to Pesticide Exposure

The intensive use of pesticides raises significant concerns regarding food safety and human health. Strawberries, often consumed without peeling, require particular attention with respect to pesticide residues. The European Directive on the sustainable use of pesticides [40] and the Common Agricultural Policy (CAP) actively promote the reduction in dependency on plant protection products by encouraging the adoption of integrated and sustainable farming practices.

In the European Union, the toxicological parameters used for pesticide risk assessment ADI, ARfD, and MRL (Maximum Residue Level) are established by EFSA in collaboration with the European Commission and Member States, as outlined in Regulation (EC) No. 396/2005 [41] and Regulation (EC) No. 1107/2009 [42].

The ADI and ARfD are based on the identification of the NOAEL, derived from toxicological studies. The MRL represents the maximum legally permitted concentration of a pesticide residue in food or feed. It is determined based on residue levels found in agronomic trials conducted under Good Agricultural Practices (GAP). A proposed MRL value is considered acceptable only if the estimated exposure remains within the safety limits defined by the ADI and ARfD. Following scientific assessment by EFSA, the MRL is officially adopted through a legislative act of the European Commission, subject to the favorable opinion of the competent Standing Committee. These limits are periodically updated to reflect new scientific evidence and to ensure public health protection.

Appendix A reports the regulatory reference values for all active substances detected through multiresidue analysis of the strawberry samples examined. A considerable variability in the established values of MRL, ARfD, and ADI can be observed among different active substances, reflecting their distinct toxicological properties and risk profiles. For instance, fenhexamid, hexythiazox, and boscalid show relatively high MRLs (10 mg kg^−1^, 6 mg kg^−1^, and 6 mg kg^−1^, respectively), while others such as emamectin and cyflufenamid have very low MRLs (0.05 mg kg^−1^ and 0.04 mg kg^−1^, respectively). As for ARfD values, high thresholds (e.g., 1 mg kg^−1^ bw for spinetoram and 0.9 mg kg^−1^ bw for captan) indicate lower concern regarding acute effects, whereas very low values (e.g., 0.005 mg kg^−1^ bw for lambda-cyhalothrin and acetamiprid) highlight greater attention to short-term toxicity. As noted in Section 2.5.1., some active substances do not have an assigned ARfD, indicating that an acute reference dose is not considered applicable. The ADI, which assesses chronic exposure, also spans a wide range of values, with upper limits such as 1.56 mg kg^−1^ bw/day for chlorantraniliprole, and very low limits such as 0.004 mg kg^−1^ bw/day for tetraconazole, 0.005 mg kg^−1^ bw/day for acetamiprid, 0.0025 mg kg^−1^ bw/day for lambda-cyhalothrin, and 0.0005 mg kg^−1^ bw/day for emamectin.

Based on the aforementioned regulations and on Regulation (EC) No. 396/2005 and its subsequent amendments, the compliance of pesticide residues detected in the analyzed food samples was assessed, with particular attention to adherence to the MRLs listed in Table 2, in order to evaluate conformity and estimate the potential risk to human health.

The data show that, with few exceptions, the detected residue concentrations remain within the MRLs established by Regulation (EC) No. 396/2005 [41].

Below are the samples that were found to contain at least one pesticide exceeding 100% of the respective MRL, and were therefore deemed non-compliant:Sample A5: Presence of 5 residues, including Flupyradifurone with an MRL exceedance of 170%;Sample S2: Presence of 3 residues, including Cyflumetofen with an MRL exceedance of 105%;Sample V1: Presence of 5 residues, including Cyflumetofen with an MRL exceedance of 123%;Sample I4: Presence of 6 residues, including Spirotetramat with an MRL exceedance of 113%.

Cyflumetofen was not only the most frequently detected residue, as reported in Section 3., but in two cases, its concentrations of 0.627 mg kg^−1^ and 0.735 mg kg^−1^ corresponded to 105% and 123% of the established MRL of 0.6 mg kg^−1^, respectively.

Some substances were detected at concentrations so low that their levels ranged from 0% to 2% of the respective MRLs; among these were hexythiazox, boscalid, fenhexamid, and chlorantraniliprole. Azoxystrobin was detected only in trace amounts in a single sample, with an MRL compliance rate of 0%.

#### 3.2.1. Results of the Acute Exposure Assessment

The IESTI calculation method used to assess the acute dietary exposure risk to pesticides is described below. As an explanatory example, the active substance flupyradifurone, detected at a concentration of 0.68 mg kg^−1^ in sample A5, was selected.

IESTI Case 1 formula was applied, with the following parameters:-HR: 0.68 mg kg^−1^ (Highest Residue found in the sample)-LP: 166.70 g/day (high consumption level for children aged 8–20 months, calculated by multiplying the standard value of 0.48 g/kg bw)-PF: 1 (Processing Factor, equal to 1 for strawberries, which are consumed raw)-CF: 1 (Conversion Factor, equal to 1 for flupyradifurone)-BW: 10.20 kg (average body weight for children aged 8–20 months)

Substituting into the formula:(5)$$\mathrm{IESTI}\;=\;\mathrm{HR}\;\times \;\mathrm{LP}\;\times \;\mathrm{PF}\;\times \;\mathrm{CF}/\mathrm{BW}\;=\;0.68\;\times \;0.1667\;\times \;1\;\times \;1/10.20\;=0.0111\;\mathrm{mg}\;{\mathrm{kg}}^{-1}\;\mathrm{bw}$$

This value was compared to the ARfD for flupyradifurone, which is 0.15 mg kg^−1^ bw:IESTI/ARfD = 0.0111/0.15 = 7.4%(6)

Using the IESTI model with a high food intake scenario (166.7 g/day) and an average body weight of 10.2 kg for the target group (children aged 8–20 months), the estimated acute exposure was 0.0111 mg kg^−1^ bw, corresponding to 7.41% of the ARfD. This result indicates that there is no acute toxicological risk for the considered population group, even under realistic worst-case dietary exposure conditions. Table 3 reports the results of active substance residues in the analyzed samples and their respective ARfD percentages (%ARfD). Regarding the substances for which acute toxicity impact on the human body can be assessed, all residues comply with the established safety limits, remaining well below the ARfD. The highest value observed was 10.87%, calculated for the tebufenpyrad residue found in sample D3. Most active substances, such as fluopyram, spinetoram, penthiopyrad, penconazole, captan, trifloxystrobin, and cyflufenamid, did not even reach 1% of the calculated ARfD, confirming a negligible risk of acute toxicity for the consumer. Particular attention is warranted for the pesticides flupyradifurone and spirotetramat, which exceeded the MRL in some samples by 170% and 113%, respectively, yet still exhibited acceptable levels in terms of acute toxicity. Flupyradifurone reached 7.40% of the reference ARfD, ranking as the second-highest value recorded, while spirotetramat showed only 0.56% of the ARfD, remaining well below the threshold of concern.

#### 3.2.2. Results of the Chronic Exposure Assessment

The following section outlines the NEDI calculation method used for the assessment of chronic dietary exposure to pesticides. As an illustrative example, the calculation model is applied to cyflumetofen, detected at an average concentration of 0.112 mg kg^−1^.

The NEDI formula includes the following parameters:(1)APR (Average Pesticide Residue): 0.112 mg kg^−1^(2)Average strawberry consumption (NL toddler): 0.344 g/kg bw/day(3)BW (Body Weight): 10.20 kg, average body weight for children aged 8–20 months

By substituting these values into the formula, the result is obtained:(7)$$\mathrm{NEDI}\;=\;\mathrm{APR}\;\times \;\mathrm{average}\mathrm{daily}\mathrm{consumption}/1000\;=\;0.112\;\times \;0.344/1000\;=0.000038528\;\mathrm{mg}\;{\mathrm{kg}}^{-1}\;\mathrm{bw}/\mathrm{die}$$

A comparison is then made with the ADI (Acceptable Daily Intake) of Cyflumetofen, set at 0.17 mg kg^−1^ bw/day.NEDI/ADI = 0.000038528/0.17 = 0.023%(8)

Applying the NEDI model, based on an APR value of 0.112 mg kg^−1^ and an average body-weight-specific consumption of 0.344 g/kg bw/day, the estimated chronic exposure to cyflumetofen for the NL toddler group is 0.0000385 mg kg^−1^ bw/day, corresponding to approximately 0.023% of the ADI (0.17 mg kg^−1^ bw/day). The resulting chronic daily exposure risk can therefore be considered negligible. Chronic dietary risk assessment allows for the evaluation of the potential impact of long-term pesticide toxicity on the human body. As shown in Table 4, all the pesticides analyzed fall within the respective ADI limits, remaining well below the acceptable daily intake threshold. The highest percentage was observed for Emamectin (15.14%), which has the lowest ADI value and thus the greatest chronic toxicity among the compounds examined.

For all other active substances, chronic exposure percentages do not reach 1%, highlighting a very high level of safety. In particular, chlorantraniliprole shows an extremely low value (0.00025%) relative to its high ADI threshold. Even for pesticides more frequently detected in the samples, such as cyflumetofen (0.023%) and bupirimate (0.087%), the exposure percentages remain very low.

Overall, the findings confirm a high level of food safety for consumers regarding the chronic consumption of strawberries.

## 4. Discussion

The presence and variation in MRLs, ARfD, and ADI for the detected active substances are crucial parameters in the management of food safety related to pesticide use, particularly for crops such as strawberries, which are consumed whole. MRLs (represent the legally permitted maximum concentrations of pesticide residues in food and are established to protect consumer health, taking into account good agricultural practices and toxicological data. Compliance with MRLs constitutes the first indicator of conformity and safety for agricultural products.

Appendix A highlights the complexity of pesticide risk assessment and underscores the importance of a rigorous monitoring and control system. Sustainable agricultural practices, such as the adoption of integrated pest management and the optimization of pesticide use, are imperative not only to minimize environmental impact but also to ensure that residue levels remain consistently below safety thresholds. Understanding these toxicological and regulatory parameters is essential for all stakeholders in the agri-food chain, from producers to consumers, in order to promote a safe and responsible food system.

ARfD and ADI provide the toxicological foundation for MRLs. For this reason, all substances analyzed for acute and chronic exposure risk assessment were within the established limits, despite the presence of some pesticides (flupyradifurone, cyflumetofen, and spirotetramat) exceeding the MRLs. It should be emphasized that the MRL is not a toxicological threshold but a legal limit, which may be 100 to 1000 times higher than the ADI or ARfD, provided that actual exposure remains below the levels considered safe, as confirmed in our study through the application of the PRIMo model to a particularly vulnerable population, taking into account overall diet and consumption frequency.

In this study, both the ARfD and the ADI attributed to two active substances commonly detected in the analyzed samples, acetamiprid and lambda-cyhalothrin, were particularly low compared to other pesticides. This finding is attributable to the neurological and systemic toxicity of these compounds demonstrated in animal experimental models, and to the precautionary principle applied by regulatory authorities to protect the most vulnerable groups, such as children.

Acetamiprid, belonging to the neonicotinoid class, acts as an agonist of nicotinic acetylcholine receptors (nAChRs), causing in insects acute effects such as hyporeactivity and tremors, and chronic effects such as behavioral and neurodevelopmental alterations [43]. Since similar risks exist for mammals, EFSA set an ARfD of 0.025 mg kg^−1^ bw, derived from an acute NOAEL identified in rodent studies and applying the usual uncertainty factor of 100 [29].

Lambda-cyhalothrin, a type II pyrethroid, is known for its high neurobehavioral toxicity, linked to interference with voltage-gated sodium channels, which leads to neuronal hyperactivity, tremors, spasms, and convulsions [44]. In children, this toxicity is even more relevant due to the greater permeability of the blood–brain barrier and the immaturity of detoxification systems [45]. Based on this evidence, its ADI was set at 0.0025 mg kg^−1^ bw and its ARfD at 0.0025 mg kg^−1^ bw, among the lowest values for pesticides authorized in Europe [44]. The presence of these active substances in the analyzed samples requires special attention in acute risk assessment, particularly regarding the risk of cumulative exposure. Fortunately, the multiresidue analysis revealed that these compounds were used sporadically (acetamiprid detected in three samples and lambda-cyhalothrin only in sample A7), resulting in an acute toxicity risk below 4% of the ARfD and a low chronic toxicity risk (0.13% of the ADI for acetamiprid and 0.21% for lambda-cyhalothrin).

Among the active substances with particularly low ADI values, in addition to acetamiprid and lambda-cyhalothrin, tetraconazole and emamectin benzoate were also found in the analyzed samples, in 13 and 6 cases, respectively, although at very low total concentrations. Tetraconazole, a triazole fungicide, has an ADI of 0.001 mg kg^−1^ bw [29]. This value was established due to potential hepatotoxicity observed in chronic toxicity studies and suspected endocrine and carcinogenic effects, although the latter were not conclusively demonstrated. Persistent hepatic enzyme induction observed in murine models led regulatory authorities to adopt a highly conservative limit, particularly considering the compound’s long half-life and bioaccumulation capacity [29].

Emamectin benzoate, an insecticide derived from avermectin, acts as a modulator of GABA- and glutamate-gated chloride channels, causing neuromuscular paralysis in insects. However, repeated-dose studies have shown that it can induce subclinical neurotoxicity in mammals, with effects on motor function, balance, and reflexes [29]. For this reason, the ADI was set at 0.0005 mg kg^−1^ bw, one of the lowest among currently authorized pesticides in Europe, while the ARfD was set at 0.01 mg kg^−1^ bw, further highlighting the need for precaution in cases of repeated or cumulative exposure. The comparison among these active substances shows that ADI values are determined not only by acute toxicity but also by chronic and systemic effects, often subclinical, that require wide safety margins to protect public health, particularly in children.

Despite the frequent detection of cyflumetofen in the analyzed samples, the toxicological reference values established by EFSA are relatively less stringent compared to other pesticides included in this study. Cyflumetofen, an acaricide, acts through the selective inhibition of mitochondrial complex II (succinate dehydrogenase), leading to disruption of energy production in target mites. Chronic toxicity studies in rodents revealed no relevant evidence of carcinogenicity, neurotoxicity, or reproductive toxicity [46]. Based on these data, the ADI was set at 0.03 mg kg^−1^ bw and the ARfD at 0.1 mg kg^−1^ bw, values significantly higher than those of emamectin, lambda-cyhalothrin, and tetraconazole, indicating overall lower systemic toxicity in cases of both acute and chronic oral exposure [29]. However, its frequent detection in strawberry samples even within legal limits suggests the need for continuous monitoring in the context of cumulative multiresidue risk assessment.

Another substance detected in the analyzed samples was flupyradifurone, a systemic insecticide belonging to the butenolide class, which acts as an agonist of nAChRs, similarly to neonicotinoids, but with a different chemical structure that confers greater selectivity for insect receptors. From a toxicological perspective, flupyradifurone has shown no genotoxic, carcinogenic, teratogenic, or neurotoxic effects in standard animal studies, and its acute toxicity is considered low [47]. Based on available data, EFSA established an ADI of 0.064 mg kg^−1^ bw and an ARfD of 0.25 mg kg^−1^ bw, values considerably higher than those of systemic insecticides such as acetamiprid or emamectin. These values reflect a better systemic tolerability, although careful surveillance remains necessary in cases of multiple exposure. Notably, its shared mode of action with neonicotinoids suggests that potential cumulative effects on the nervous system should be considered in multiresidue risk assessment models.

Among the detected active substances, pyrimethanil showed the highest mean, maximum, and total concentrations in the analyzed strawberry samples. Nevertheless, EFSA set its ADI at 0.17 mg kg^−1^ bw and its ARfD at 0.6 mg kg^−1^ bw, values considerably higher than those of other pesticides detected [48], indicating relatively low toxicity. Toxicological studies revealed no significant carcinogenic, mutagenic, or teratogenic effects, and chronic systemic toxicity was observed only at high doses [48]. Its low bioaccumulation potential, lack of neurotoxic effects, and rapid elimination explain why EFSA set relatively broad safety margins. However, its high detection frequency, combined with higher concentrations than other pesticides, makes pyrimethanil a compound that requires careful monitoring, especially in the context of cumulative dietary exposure and in foods intended for infants, where the toxicological safety margin is narrower due to lower body weight and higher consumption per unit body mass. Overall, the risk assessment highlights that substances with very low ADI and ARfD values, such as emamectin, lambda-cyhalothrin, and tetraconazole, although present at lower concentrations, require greater attention compared to active substances such as pyrimethanil, flupyradifurone, or cyflumetofen, which are characterized by lower systemic toxicity. This underscores the importance of an assessment framework that considers not only the detected concentrations but also the toxicological profile and the potential cumulative effects, particularly in vulnerable population groups. To provide a comprehensive interpretation of these findings, results were compared with data from studies conducted in different regions of the world. Keklik et al. [36] analyzed approximately 245 strawberry samples in Turkey, identifying pyrimethanil (around 30.2%) among the most frequently detected pesticides, consistent with this study, where pyrimethanil accounted for approximately 21.6% of detected pesticides. Chu et al. [49] in China investigated about 26 pesticides used in strawberry cultivation, with pyrimethanil (26.7%) and acetamiprid (14.7%) being the most commonly detected. El Sheikh et al. [18] in Egypt reported higher concentrations of lambda-cyhalothrin (29.3%) in the samples analyzed. Although the percentages differ, all these studies examined pesticides in common with those assessed in this paper, particularly pyrimethanil, acetamiprid, and lambda-cyhalothrin. The health risk assessment in these studies is consistent with these findings: the concentrations detected do not appear to pose a significant risk to either adult or child populations, neither in terms of acute nor chronic exposure.

This study provides valuable insights into pesticide residues in strawberries and their potential health risks, while acknowledging certain limitations. Although the overall health risk was found to be low, the detection of multiple residues in over half of the samples and occasional exceedances of MRLs indicate the need for strengthened supervision and adherence to GAP to reduce unnecessary pesticide use. The risk assessment focused on individual compounds, leaving potential additive or synergistic effects of multiple residues unaddressed; nevertheless, the study provides a foundation for future cumulative risk assessments, particularly for pesticides with shared toxicity mechanisms. Finally, monitoring was limited to harvest-stage samples assuming unwashed consumption; including market samples, household washing or processing effects, and diverse consumer groups in future studies would improve real-world relevance. Overall, despite these limitations, the study offers reliable data and a solid basis for public health assessment, regulatory guidance, and the promotion of safer agricultural practices.

## 5. Conclusions

This study confirms that locally produced strawberries from the Agro Aversano area generally comply with current European food safety standards, as the majority of pesticide residues were found within legal thresholds. However, the results also highlight that compliance with MRLs alone may not be sufficient to ensure consumer safety, particularly when considering the toxicological thresholds of certain substances and the potential cumulative effects of multiple exposures. From a broader public health perspective, these findings underscore the need to integrate toxicological reference values, frequency of detection, and cumulative risk into dietary exposure assessments in order to better protect vulnerable groups such as children. In terms of regulatory applications, the outcomes of this study provide valuable evidence to support the refinement of monitoring programs. Incorporating probabilistic and cumulative risk models, as well as updated local consumption data, could strengthen food safety surveillance and enhance the responsiveness of health authorities to emerging risks. Finally, the results reinforce the importance of promoting sustainable pest management practices in intensive agricultural areas like Agro Aversano. Greater adoption of integrated pest management strategies, such as biopesticides, resistant crop varieties, crop rotation, and the use of natural predators, represents a critical step toward reducing chemical inputs, minimizing environmental impacts, and ensuring both consumer protection and long-term agricultural sustainability.

## Figures and Tables

**Figure 1 foods-14-03470-f001:**
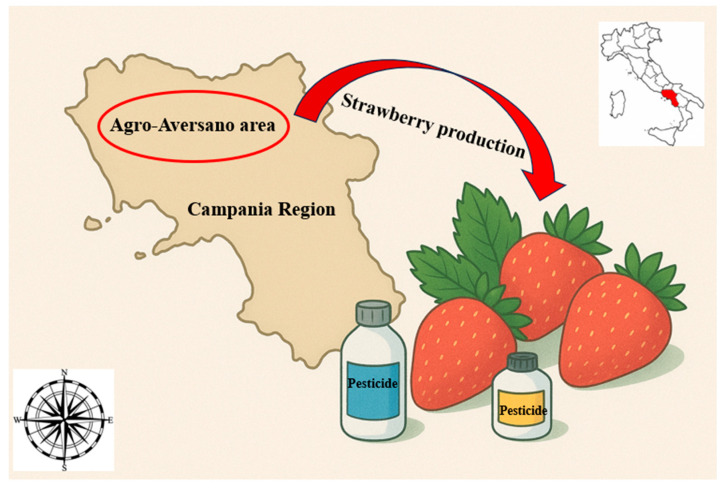
Map of the study area in the Campania Region, southern Italy.

**Figure 2 foods-14-03470-f002:**
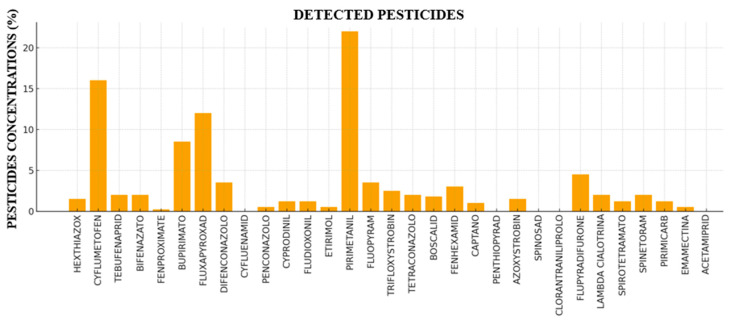
Percentage concentration of individual pesticide active substances. Trend of the percentage concentration of different pesticide active substances detected in analyzed strawberry samples.

**Figure 3 foods-14-03470-f003:**
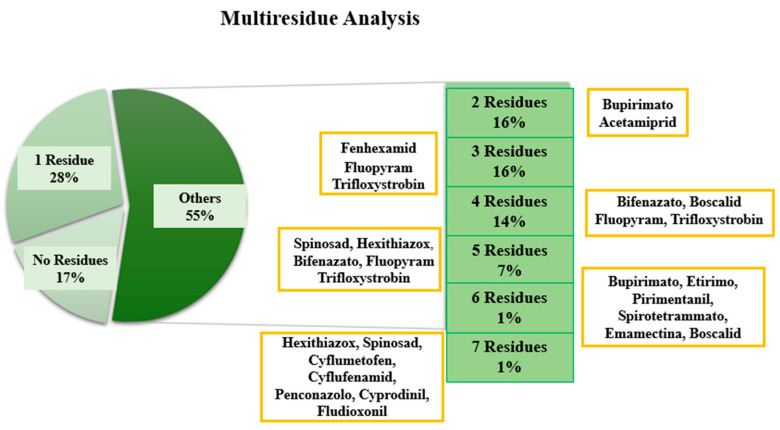
Percentage distribution of samples according to the number of detected pesticides. 17% of the samples showed no residues, while 28% contained a single residue. The remaining 55% had between two and seven residues, with progressively decreasing values.

**Table 1 foods-14-03470-t001:** Pesticide concentrations in strawberry samples. Statistical summary of the amounts of various compounds (acaricides, fungicides, insecticides), reporting for each the mean concentration (Mean), minimum (Min), maximum (Max), and total (Total).

Compound	Pesticides	Mean (mg kg^−1^)	Min (mg kg^−1^)	Max (mg kg^−1^)	Total (mg kg^−1^)
Acaricides	Hexithiazox	0.039	0.010	0.075	0.310
	Cyflumetofen	0.112	0.011	0.735	2.803
	Tebufenaprid	0.073	0.023	0.133	0.291
	Bifenazato	0.040	0.010	0.083	0.357
	Fenpyroximate	0.079	0.079	0.079	0.079
Fungicides	Bupirimato	0.126	0.012	0.350	1.508
	Fluxapyroxad	0.267	0.018	0.650	2.138
	Difenconazolo	0.115	0.010	0.250	0.576
	Cyflufenamid	0.015	0.015	0.015	0.030
	Penconazolo	0.035	0.018	0.052	0.139
	Cyprodinil	0.260	0.260	0.260	0.260
	Fludioxonil	0.125	0.030	0.220	0.250
	Etirimol	0.025	0.010	0.054	0.151
	Pirimentanil	0.634	0.011	2.000	3.806
	Fluopyram	0.065	0.012	0.200	0.720
	Trifloxystrobin	0.052	0.010	0.140	0.464
	Tetraconazolo	0.034	0.012	0.118	0.437
	Boscalid	0.059	0.012	0.130	0.296
	Fenhexamid	0.099	0.015	0.170	0.497
	Captano	0.030	0.030	0.030	0.030
	Penthiopyrad	0.124	0.021	0.226	0.247
	Azoxystrobin	0.012	0.012	0.012	0.012
Insecticides	Spinosad	0.030	0.011	0.095	0.273
	Clorantraniliprolo	0.011	0.010	0.012	0.034
	Flupyradifurone	0.195	0.020	0.680	0.780
	Lambda cialotrina	0.015	0.015	0.015	0.015
	Spirotetrammato	0.196	0.052	0.340	0.392
	Spinetoram	0.032	0.016	0.051	0.162
	Pirimicarb	0.092	0.012	0.220	0.369
	Emamectina	0.022	0.010	0.035	0.133
	Acetamiprid	0.019	0.012	0.033	0.058

**Table 2 foods-14-03470-t002:** Maximum Residue Limits (MRLs), Acute Reference Doses (ARfDs), and Acceptable Daily Intakes (ADIs) for pesticides detected in strawberries, including individual residue levels, compliance with MRLs, and EU regulatory parameters to support food safety assurance. mg kg^−1^. The - indicates an unavailable value.

Pesticides	Results	MRL (mg kg^−1^)	MRL%	ARfD (mg kg^−1^-bw)	ADI (mg kg^−1^ bw/Day)
Bupirimate	0.054	1.5	4%	-	0.05
	0.130		9%		
	0.170		11%		
	0.270		18%		
	0.055		4%		
	0.062		4%		
	0.350		23%		
	0.012		1%		
	0.220		15%		
	0.075		5%		
	0.096		6%		
	0.014		1%		
Spinosad	0.095	0.3	32%	-	0.024
	0.012		4%		
	0.086		29%		
	0.015		5%		
	0.011		4%		
	0.018		6%		
Chlorantraniliprole	0.010	1	1%	-	1.56
	0.012		1%		
Flupyradifurone	0.680	0.4	170%	0.15	0.064
	0.020		5%		
	0.060		15%		
Fluxapyroxad	0.018	4	0%	0.25	0.02
	0.030		1%		
	0.200		5%		
	0.480		12%		
	0.500		13%		
	0.650		16%		
	0.110		3%		
	0.150		4%		
Difenoconazole	0.010	2	1%	0.16	0.01
	0.250		13%		
	0.220		11%		
	0.026		1%		
	0.070		4%		
Hexythiazox	0.037	6	1%	-	0.03
	0.075		1%		
	0.050		1%		
	0.020		0%		
	0.010		0%		
	0.072		1%		
	0.034		1%		
	0.012		0%		
Lambda-cyhalothrin	0.015	0.2	8%	0.005	0.0025
Cyflumetofen	0.023	0.6	4%	-	0.17
	0.012		2%		
	0.013		2%		
	0.130		22%		
	0.100		17%		
	0.011		2%		
	0.627		105%		
	0.735		123%		
	0.063		11%		
	0.224		37%		
	0.070		12%		
	0.024		4%		
	0.034		6%		
	0.256		43%		
	0.014		2%		
Cyflufenamid	0.015	0.04	38%	0.05	0.04
Penconazole	0.047	0.5	9%	0.5	0.03
	0.022		4%		
	0.052		10%		
	0.018		4%		
Cyprodinil	0.260	5	5%	-	0.03
Fludioxonil	0.220	4	6%	-	0.37
	0.030		1%		
Ethirimol	0.017	0.3	6%	Not applicable	Not applicable
	0.020		7%		
	0.010		3%		
	0.054		18%		
	0.040		13%		
Pyrimethanil	0.160	5	3%	-	0.17
	2.000		40%		
	1.200		24%		
	0.350		7%		
	0.085		2%		
	0.011		0%		
Fluopyram	0.100	2	5%	0.5	0.012
	0.013		1%		
	0.031		2%		
	0.035		2%		
	0.077		4%		
	0.116		6%		
	0.200		10%		
	0.012		1%		
	0.110		6%		
	0.014		1%		
Tebufenpyrad	0.023	1	2%	0.02	0.01
	0.030		3%		
	0.133		13%		
	0.105		11%		
Trifloxystrobin	0.065	1	7%	0.5	0.1
	0.010		1%		
	0.025		3%		
	0.035		4%		
	0.066		7%		
	0.140		14%		
	0.100		10%		
	0.013		1%		
Bifenazate	0.083	3	3%	0.1	0.01
	0.080		3%		
	0.072		2%		
	0.012		0%		
	0.050		2%		
	0.026		1%		
	0.010		0%		
Spirotetramat	0.340	0.3	113%	1	0.05
	0.052		17%		
Fenpyroximate	0.079	0.3	26%	-	0.2
Tetraconazole	0.060	0.15	40%	0.05	0.004
	0.016		11%		
	0.012		8%		
	0.118		79%		
	0.100		67%		
	0.025		17%		
	0.016		11%		
	0.018		12%		
	0.013		9%		
Boscalid	0.012	6	0%	-	0.04
	0.080		1%		
	0.130		2%		
	0.054		1%		
	0.020		0%		
Fenhexamid	0.015	10	0%	-	0.2
	0.170		2%		
	0.051		1%		
	0.100		1%		
	0.161		2%		
Spinetoram	0.051	0.2	26%	0.1	0.025
	0.017		9%		
	0.016		8%		
	0.039		20%		
Captan	0.030	1.5	2%	0.9	0.25
Penthiopyrad	0.226	3	8%	0.75	0.1
	0.021		1%		
Pirimicarb	0.220	1.5	15%	0.1	0.035
	0.090		6%		
	0.047		3%		
	0.012		1%		
Azoxystrobin	0.012	10	0%	-	0.2
Emamectin	0.013	0.05	26%	0.01	0.0005
	0.010		20%		
	0.035		70%		
	0.031		62%		
Acetamiprid	0.012	0.5	2%	0.005	0.005
	0.033		7%		
	0.013		3%		

**Table 3 foods-14-03470-t003:** Pesticide residues in analyzed samples and corresponding ARfD% values. Reported are residue levels of each pesticide, MRLs (mg kg^−1^), ARfD (mg kg^−1^ bw), and the percentage of the concentration relative to the ARfD (ARfD%).

Pesticides	Results	ARfD (mg kg^−1^-bw)	ARfD%
Flupyradifurone	0.68	0.15 mg kg^−1^	7.40
Fluxapyroxad	0.65	0.25 mg kg^−1^	4.25
Difenoconazole	0.25	0.16 mg kg^−1^	2.55
Lambda-cyhalothrin	0.015	0.005 mg kg^−1^	4.90
Cyflufenamid	0.015	0.05 mg kg^−1^	0.49
Penconazole	0.052	0.5 mg kg^−1^	0.17
Fluopyram	0.2	0.5 mg kg^−1^	0.65
Tebufenpyrad	0.133	0.02 mg kg^−1^	10.87
Trifloxystrobin	0.14	0.5 mg kg^−1^	0.46
Bifenazate	0.083	0.1 mg kg^−1^	1.36
Spirotetramat	0.34	1 mg kg^−1^	0.56
Fenpyroximate	0.079	0.02 mg kg^−1^	6.46
Tetraconazole	0.118	0.05 mg kg^−1^	3.86
Spinetoram	0.051	0.1 mg kg^−1^	0.83
Captan	0.03	0.9 mg kg^−1^	0.05
Penthiopyrad	0.226	0.75 mg kg^−1^	0.49
Pirimicarb	0.22	0.1 mg kg^−1^	3.60
Emamectin	0.035	0.01 mg kg^−1^	5.72
Acetamiprid	0.013	0.005 mg kg^−1^	4.25

**Table 4 foods-14-03470-t004:** Estimated chronic exposure to pesticide residues in strawberries. Assessment of the percentage of the ADI (%ADI) for 27 detected pesticides, based on the average residue concentration (APR), daily intake per kilogram of body weight (0.344 g/kg bw/day), and the reference body weight of 10.2 kg (NL toddler category, according to the EFSA PRIMo rev. 3.1 model).

Principi Attivi	Risultato Medio (mg kg^−1^)	ADI (mg kg^−1^-bw/Die)	ADI%
Cyflumetofen	0.112	0.17	0.023
Tebufenpyrad	0.073	0.010	0.25
Bifenazate	0.040	0.010	0.14
Fenpyroximate	0.079	0.010	0.27
Bupirimato	0.126	0.050	0.087
Fluxapyroxad	0.267	0.020	0.46
Difenoconazole	0.115	0.010	0.40
Cyflufenamid	0.015	0.040	0.013
Penconazole	0.035	0.030	0.04
Cyprodinil	0.260	0.030	0.30
Fludioxonil	0.125	0.370	0.012
Etirimol	0.025	0.035	0.025
Pyrimethanil	0.634	0.170	0.13
Fluopyram	0.065	0.012	0.19
Trifloxystrobin	0.052	0.1	0.018
Tetraconazole	0.034	0.004	0.30
Boscalid	0.059	0.040	0.051
Fenhexamid	0.099	0.200	0.017
Captan	0.030	0.25	0.004
Penthiopyrad	0.124	0.1	0.04
Azoxystrobin	0.012	0.200	0.0021
Spinosad	0.030	0.025	0.041
Clorantraniliprolo	0.011	1.565	0.00025
Flupyradifurone	0.195	0.064	0.10
Lambda-cyhalothrin	0.015	0.0025	0.21
Spirotetramat	0.196	0.050	0.13
Spinetoram	0.032	0.025	0.04
Pirimicarb	0.092	0.035	0.09
Emamectin	0.022	0.0005	15.136
Acetamiprid	0.019	0.005	0.13
Hexithiazox	0.039	0.030	0.04

## Data Availability

All related data and methods are presented in this manuscript and Appendix A. Additional inquiries should be addressed to the corresponding author.

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
