# Peer review of "Analysis and Risk Assessment of Pesticide Residues in Strawberry Using the PRIMo Model: Detection, Public Health and Safety Implications"

_foods, 2025, doi:10.3390/foods14203470_

Round 1

Reviewer 1 Report

Comments and Suggestions for Authors

This manuscript combines QuEChERS and LC-MS/MS to establish a detection method for 850 pesticide residues in strawberries, ands this method was used to determine the pesticide residue levels in 83 samples from the Agro Aversano area. The results showed that each sample contained at least 31 different active substances. Based on the pesticide residues present in the sample, combined with the EFSA PRIMo model (v.3.1), the dietary risk assessment was conducted on the NL toddler subgroup, and the results showed that acute and chronic exposures were below toxicological reference values. The research results provide theoretical support for the quality and safety of strawberries.

  • The abbreviation format used in the manuscript should be consistent. For example, Acute Reference Dose (ARfD).And the abbreviation needs to be written in its full name for the first time, but it is not necessary in the future, such as. LOQs in 150 line. Please check the abbreviations in the manuscript.
  • The Method section description is lengthy, such as line 188-198 should not be included in the Method.
  • The paragraph format is confusing. Please unify the paragraph format in the manuscript.
  • The manuscript needs to be concise.

Reviewer 2 Report

Comments and Suggestions for Authors

This study analyzed pesticide residues in 83 strawberry samples from the Agro Aversano region of Italy. A total of 31 pesticides were detected, primarily fungicides, with multiple residues being a common occurrence (55% of samples contained more than two pesticides). Although some samples exceeded the EU's Maximum Residue Levels (MRLs), a risk assessment conducted using the EFSA PRIMo model for the most sensitive population (young children) indicated that both acute and chronic exposure levels for all pesticides were significantly below the safety thresholds (ARfD and ADI), suggesting a negligible health risk. Although the methodology of this study is relatively straightforward, it reveals the current status of pesticide residues in strawberries from a specific region and provides a highly valuable risk assessment. The findings have significant public education value in helping the general public understand pesticide residues and offer practical guidance for the scientific application of pesticides in strawberry cultivation. However, the study also has the following limitations:

  1. Strengthen supervision and guidance for the management of pesticide use at the source.

The study found that despite the low health risk, the phenomenon of multiple pesticide residues (55% of samples contained 2 or more pesticides) and exceedances of Maximum Residue Levels (MRLs) in individual samples still exist. This suggests that in the agricultural production stage, there may be issues such as non-standard pesticide use, failure to strictly adhere to pre-harvest intervals, or overuse. Therefore, it is necessary to enhance technical guidance and supervision for growers, strictly promote Good Agricultural Practices (GAP), and reduce unnecessary pesticide use and the occurrence of multiple residues at the source.

  1. Promote the establishment and application of cumulative risk assessment methods for mixed residues.

The paper clearly points out that current regulations primarily set limits based on the toxicity of individual substances, while the potential "additive effects" or "synergistic effects" of multiple pesticide residues are a concern. Although this study mentions this risk, the assessment is still based on individual substances. Future efforts should prioritize promoting the establishment of scientific cumulative risk assessment models, especially for pesticides with common mechanisms of toxicity (e.g., triazole fungicides), to enable a more comprehensive assessment of their long-term health effects, particularly for sensitive populations like children.

  1. Expand the monitoring scope to better reflect actual consumer exposure.

    This study focused on samples at the harvest stage from the production area and assumed strawberries were consumed "unwashed" for risk assessment. To better reflect real consumption scenarios, future monitoring and research should include samples from market circulation channels and systematically evaluate the actual reduction effects of household washing and processing (e.g., making jam) on pesticide residues. Furthermore, the risk assessment could be extended to more consumer groups (e.g., adults in different regions) and broader dietary patterns, making the assessment results more universally applicable and providing a more precise basis for public policy.

4.Line 597: "3.3 Discussions" should be "4. Discussions".

5.Lines 707-737: The conclusion section could be more concise and condensed, stating the findings directly.

6.It is recommended that the authors provide a more detailed description of the sample processing methods and steps to facilitate the replication (and learning) of the experimental procedure by readers. Specifically, the characteristic ion peaks for the qualitative and quantitative analysis of each pesticide should be clearly stated.

7.The supporting materials are relatively sparse; supplementary materials, methods, and supporting documentation related to the experimental process are insufficient. It is recommended to include relevant chromatograms and mass spectra in the supporting materials.

Reviewer 3 Report

Comments and Suggestions for Authors
  1. There is no list of compounds to be determined, validation parameters, or chromatographic parameters. It is unclear which compounds the authors are determining and what validation parameters they obtained (whether the method is suitable for determining so many pesticides). Please supplement this information.
  2. No discussion with other researchers. Did they detect similar compounds, did they identify risks? (point 3.3)
  3. Figure 2 is unclear. Why are the points on the axis connected? Is there a correlation between them? On the y-axis, is the concentration of pesticides expressed as a percentage? Please correct and clarify.
  4. Figure 3 - instead of 0 residues, no residues. A little more about multiresidues for example, which combinations occurred most frequently?
  5. What do E5 (line 392), M3 (line 406), and similar symbols mean in the text?
  6. Table 2 and Table S1 need to be corrected. NDPs are repeated in both tables. In Table 2, the repetition of units and NDPs in columns is unnecessary. Merge the two tables.
  7. Table 3 in columns unnecessary repeating units and %, if the first row explains the units in which the parameters are given.
  8. For example, line 600 – the abbreviation MRL has already been explained twice (lines 436, 442). An abbreviation should be explained once when it first appears; check all abbreviations in the manuscript.

Reviewer 4 Report

Comments and Suggestions for Authors

The manuscript entitled ,,Analysis and Risk Assessment of Pesticide Residues in Strawberry using the PRIMo Model: Detection, Public Health and Safety Implications" aims to investigate the detection of pesticide residues in strawberries at harvest and evaluate the associated health risk. The topic of this work is important and relevant, as strawberries are widely consumed and frequently reported to contain pesticide residues. The manuscript is well-written, structured, and within the scope of the journal. My comments are given below.

The introduction is clearly presented, but it might be slightly expanded with additional context and references, including similar studies on pesticide residues in strawberries from other countries. Also, the aim of the study should be complemented with a clear statement of novelty.

In section 2. Materials and Methods, the authors should consider including the representativeness of the sampling (number of farms, distribution across the region, and seasonality), which should be discussed in more detail.

Throughout the manuscript, the full names of abbreviations should be provided before their first use, and care should be taken to avoid reintroducing the same abbreviation multiple times.

Units should be written uniformly (e.g., consistently as mg/kg or mg kg⁻¹), and the chemical names of pesticides should not begin with capital letters unless they are placed at the start of a sentence.

The quality of the figures should be improved to enhance readability and interpretation, while in line 393 dashes should be replaced with commas.

In Table 2, the red highlighting of the MRL% column for flupyradifurone, cyflumetofen, and spirotetramat should be removed, as it unnecessarily distracts from the data presentation. In Table 3, the authors should avoid writing the units in each cell of the column.

The conclusion section (Section 5) currently appears as a summary of findings. It could be strengthened by highlighting the broader public health implications, potential applications for regulatory monitoring, and recommendations for sustainable pest management practices. Also, the last paragraph should avoid starting with "In conclusion," since the entire section already serves that purpose. Please consider changing this.

Round 2

Reviewer 3 Report

Comments and Suggestions for Authors

No comments